# Repeatability of measuring the vessel density in patients with retinal vein occlusion: An optical coherence tomography angiography study

**Kyeung-Min Kim[1], Min-Woo Lee[1,2], Hyung-Bin Lim[1], Hyung-Moon Koo[1], Yong-Il Shin[1], Jung-Yeul Kim[1]** *

1 Department of Ophthalmology, Chungnam National University College of Medicine, Daejeon, Republic of Korea, 2 Department of Ophthalmology, Konyang University College of Medicine, Daejeon, Republic of Korea

* kimjy@cnu.ac.kr

## Abstract

### Purpose

To determine the repeatability of superficial vessel density measurements using Spectral domain Ocular coherence tomography angiography(SD-OCTA) in patients diagnosed with retinal vein occlusion(RVO).

### Design

Prospective observational study.

### Subjects

Patients who visited our retinal clinic from August 2017 to August 2018, diagnosed with RVO were recruited for the study.

### Methods

Two consecutive 3×3 mm pattern scans were performed using the Cirrus HD-OCT 5000 along with AngioPlex software (Carl Zeiss Meditec) in each eye by single skilled examiner. All scans were analyzed using en face OCTA images to measure vessel density (VD) automatically. For further analysis of the effect of central macular thickness(CMT), eyes were divided into two groups according to CMT of 400μm (Group 1: CMT > 400μm, Group 2: CMT < 400μm). To identify factors affecting the repeatability of VD measurements, linear regression analyses were conducted for the coefficient of variation (CV) of VD by investigating demographics and ocular variables.

### Main outcome measures

The intraclass correlation coefficient (ICC), coefficient of variation (CV) of VD measurements.

**Data Availability Statement:** All relevant data are within the paper and its Supporting Information files.

**Funding:** The author(s) received no specific funding for this work.

**Competing interests:** The authors have declared that no competing interests exist.

## Results

A total of 57 eyes from 57 patients were examined: 35 eyes with BRVO and 22 eyes with CRVO. In all 57 eyes with RVO, the ICC and CV of the full VD(VD of 3mm diameter circle) were 0.800 and 10.61%, respectively. Univariate analyses showed that the mean CMT (B, 0.001; p<0.001) and mean ganglion cell-Inner plexiform layer (GC-IPL) thickness (B, −0.002; p = 0.020) were significant factors that affected the repeatability. Multivariate analyses of these two factors showed that only mean CMT was a significant factor. The ICC and CV of the full VD in group 1 (CMT > 400μm) were 0.348 and 22.55% respectively. In group 2 (CMT < 400μm), the ICC and CV of the full VD were 0.910 and 7.76%, respectively.

## Conclusions

The repeatability of VD measurement in eyes with RVO was reasonably comparable to previous studies. Repeatability of VD measurement was significantly affected by central macular thickness.

## Introduction

Retinal vein occlusion (RVO) is the second most common retinal vascular disease after diabetic retinopathy. Nearly 2% of people over 40 years of age are diagnosed with RVO [1, 2]. RVO can cause various degrees of vision problems in the older worldwide population. Given the prevalence of the aging population throughout the world, the accurate diagnosis and adequate management of RVO have become critically important [3–6].

Fluorescein angiography (FA) has been the method primarily chosen for assessing vascular abnormalities associated with RVO. However, FA has many limitations because it is a time-consuming and invasive procedure, and the diffusion of dye makes it difficult to observe the microvasculature in the late phase of the exam. In contrast, optical coherence tomography angiography (OCTA) is noninvasive and less time-consuming, and it provides depth-resolved images to visualize the retinal vasculature in multiple layers. OCTA also provides quantitative metrics of the retinal microvasculature such as vessel density (VD), perfusion density, and the foveal avascular zone (FAZ) area of the retinal capillary plexus [7–11]. Even though OCTA has several limitations such as projection artifacts and narrow field of view, this novel technique can provide clinicians with microvascular information that can assist diagnosis and treatment of many types of retinal vascular diseases [12, 13].

Many studies reported microvascular changes in eyes with RVO using OCTA [3,14,15]. At the same time, the reliability and efficacy of the OCTA have been questioned. The repeatability of this new device have been reported in many studies in normal eyes. However, there is limited study on the repeatability of OCTA in eyes with RVO. In the present study, we determined the repeatability of VD measurements using the Cirrus HD-OCT 5000 with AngioPlex software (version 10.0; Carl Zeiss Meditec, Dublin, CA, USA) in patients diagnosed with RVO. We also identified factors affecting the repeatability of VD measurements.

## Materials and methods

### Patients

This study adhered to the tenets of the Declaration of Helsinki and was approved by the Institutional Review Board of Chungnam National University Hospital (Daejeon, Republic of

Korea). The study was performed prospectively using patients who were diagnosed with RVO who visited our retinal clinic from August 2017 to August 2018. Patients were diagnosed with RVO through comprehensive examinations including funduscopy, OCT and FA. Patients were excluded if they had any intraocular surgery except for cataract surgery. Each patient gave a detailed history and underwent testing for best-corrected visual acuity (BCVA), intraocular pressure using noncontact tonometry, spherical equivalent, axial length using an IOL Master (Carl Zeiss, Jena, Germany), and the mean central macular thickness (CMT) and ganglion cell-inner plexiform layer (GC-IPL) thickness using a Cirrus HD-OCT 5000 (Carl Zeiss Meditec).

## OCTA measurements

Two consecutive measurements were performed by a single skilled examiner using the Cirrus HD-OCT 5000 along with AngioPlex software (Carl Zeiss Meditec). The AngioPlex used a center wavelength of 840 nm, taking 68,000 A-scans/s to obtain high-resolution microvascular images. The instrument is based on the optical microangiography (OMAG) algorithm, and the retinal tracking program helps provide high sensitivity and accuracy. We used a $3 \times 3$ mm pattern scan to measure the central foveal area, and all scans were analyzed using en face OCTA images generated automatically by the OMAG algorithm used in the AngioPlex software. The VD (defined as the total length of perfused vasculature per unit area in the region of measurement) of the superficial layer was measured automatically by the software, which quantitated the VD of a local region of tissue according to the Early Treatment of Diabetic Retinopathy Study (ETDRS) subfields. The superficial layer was defined as the layer starting from the internal limiting membrane (ILM) to the inner plexiform layer (IPL). The IPL boundary was calculated as 70% of the distance from the ILM to the estimated boundary of the outer plexiform layer, which was determined to be 110 μm above the retinal pigment epithelium boundary as automatically detected by the software. AngioPlex software also measures the foveal avascular zone(FAZ) metrics automatically. It provides the measurement of FAZ area and FAZ perimeter. To analyze the repeatability of VD measurements using the automatic AngioPlex software, we did not make any manual adjustments. In this study, we analyzed the VD of the full area (3 mm diameter ring), inner area (1mm diameter inner ring), and each sector of ETDRS subfields. We also analyzed the FAZ area and FAZ perimeter. We excluded OCTA images with a signal strength $< 7$.

## Statistical analyses

To analyze the repeatability of the VD in patients with RVO, we calculated the intraclass correlation coefficient (ICC), coefficient of variation (CV), and test-retest standard deviation (TRTSD) of the full VD. An ICC (the ratio of the subject variance to the total variance) close to 1 meant that the variance was low in the same examination. The CV (%) was calculated as $100 \times SD/$overall mean, and a value $< 10\%$ meant good repeatability of the measurement. TRTSD was calculated as the square root of the within-subject mean square for error. The agreement between two measurements was evaluated using Bland-Altman plots. To identify factors affecting the repeatability of VD measurements, linear regression analyses were conducted for the CV of the full VD by investigating demographic and ocular variables. Multi-variate analyses were performed for significant values of $p < 0.05$ obtained by univariate analyses.

## Results

### Demographics and main criteria

A total of 62 eyes from 62 patients were recruited. 5 patients were excluded for poor image quality (signal strength $< 7$). Finally, 57 eyes from 57 patients were examined: 35 eyes with

**Table 1. Demographics and baseline characteristics of patients.**

| | |
|---|---|
| Number of subjects | 57 |
| Branch retinal vein occlusion | 35 |
| Central retinal vein occlusion | 22 |
| Age (years, mean±SD) | 65.6±9.1 |
| Male gender (%) | 42.1 |
| Diabetes mellitus (%) | 24.6 |
| Hypertension (%) | 42.1 |
| Right laterality (%) | 45.6 |
| Phakic eye (%) | 80.7% |
| BCVA (logMAR, mean±SD) | 0.34±0.34 |
| SE (dioptres, mean±SD) | +0.14±1.69 |
| IOP (mmHg, mean±SD) | 16.3±2.8 |
| Axial length (mm, mean±SD) | 23.5±0.8 |
| Mean signal strength (mean±SD) | 8.6±1.2 |
| Mean CMT (μm, mean±SD) | 353.7±139.4 |
| Mean GC-IPL thickness (μm, mean±SD) | 53.7±29.2 |

BCVA, best corrected visual acuity; SE, spherical equivalent; IOP, intraocular pressure; CMT, central macular thickness; GC-IPL, ganglion cell-inner plexiform layer.

BRVO and 22 eyes with CRVO. The mean age was 65.6 years, the mean BCVA was 0.34 the mean spherical equivalent was +0.14, the mean axial length was 25.5 mm, the mean CMT was 353.7 μm, and the mean GC-IPL thick-ness was 53.7 μm (Table 1). Baseline characteristics of subgroups divided according to CMT of 400 μm is shown in Table 2. There were no significant differences between the two groups except for mean CMT.

## Repeatability of the VD in RVO patients

In all 57 eyes with RVO, the ICC and CV of the full VD were 0.800 and 10.61%, respectively. In each sector, the ICC of the center, superior, nasal, inferior, and temporal areas were 0.755, 0.843, 0.824, 0.855, and 0.758, respectively, and the CV of the center, superior, nasal, inferior,

**Table 2. Demographics and baseline characteristics of each group.**

| | Group 1 (CMT>400) | Group 2 (CMT<400) |
|---|---|---|
| Number of subjects | 16 | 41 |
| Age (years, mean±SD) | 64.8±9.7 | 66.0±8.9 |
| BCVA (logMAR, mean±SD) | 0.39±0.28 | 0.31±0.36 |
| SE (dioptres, mean±SD) | +0.31±2.19 | +0.07±1.49 |
| IOP (mmHg, mean±SD) | 16.6±2.9 | 16.2±2.8 |
| Axial length (mm, mean±SD) | 23.4±0.9 | 23.6±0.8 |
| Mean signal strength (mean±SD) | 8.1±1.2 | 8.8±1.1 |
| Mean CMT (μm, mean±SD) | 544.7±105.4 | 279.1±52.5 |
| Mean GC-IPL thickness (μm, mean±SD) | 20.1±12.9 | 66.9±22.3 |

Group 1: Eyes with central macular thickness greater than 400 μm; Group 2: Eyes with central macular thickness lesser than 400 μm.

BCVA, best corrected visual acuity; SE, spherical equivalent; IOP, intraocular pressure; CMT, central macular thickness; GC-IPL, ganglion cell-inner plexiform layer.

**Table 3. First and second mean values, intraclass correlation coefficient, coefficient of variation and test-retest standard deviation of vessel density in patients with RVO.**

|  | First mean VD | Second mean VD | ICC | CV (%) | TRTSD |
|---|---|---|---|---|---|
| Full | 15.74±4.2 | 16.02±3.9 | 0.800 | 10.61 | 1.06 |
| Inner | 16.83±4.3 | 17.21±3.9 | 0.852 | 8.28 | 0.89 |
| Sectorial |  |  |  |  |  |
| Central | 8.3±3.6 | 8.7±3.7 | 0.755 | 21.48 | 1.09 |
| Superior | 17.0±5.0 | 16.9±4.4 | 0.843 | 10.30 | 1.08 |
| Nasal | 17.7±4.8 | 17.9±4.7 | 0.824 | 10.59 | 1.06 |
| Inferior | 15.9±5.4 | 16.5±5.0 | 0.855 | 11.88 | 1.15 |
| Temporal | 16.6±4.7 | 17.7±4.1 | 0.758 | 11.17 | 1.20 |

CV, coefficient of variation; ICC, intraclass correlation coefficient; TRTSD, test-retest standard deviation; VD, vessel density.

and temporal areas were 21.48%, 10.30%, 10.59%, 11.88%, and 11.17%, respectively (Table 3). Using a Bland-Altman plot, the difference was close to zero in the measurement of VD (Fig 1).

In FAZ analysis, we had to exclude 20 eyes which AngioPlex software could not detect the FAZ. Then we reviewed the FAZ line that was drawn automatically by the Angioplex software

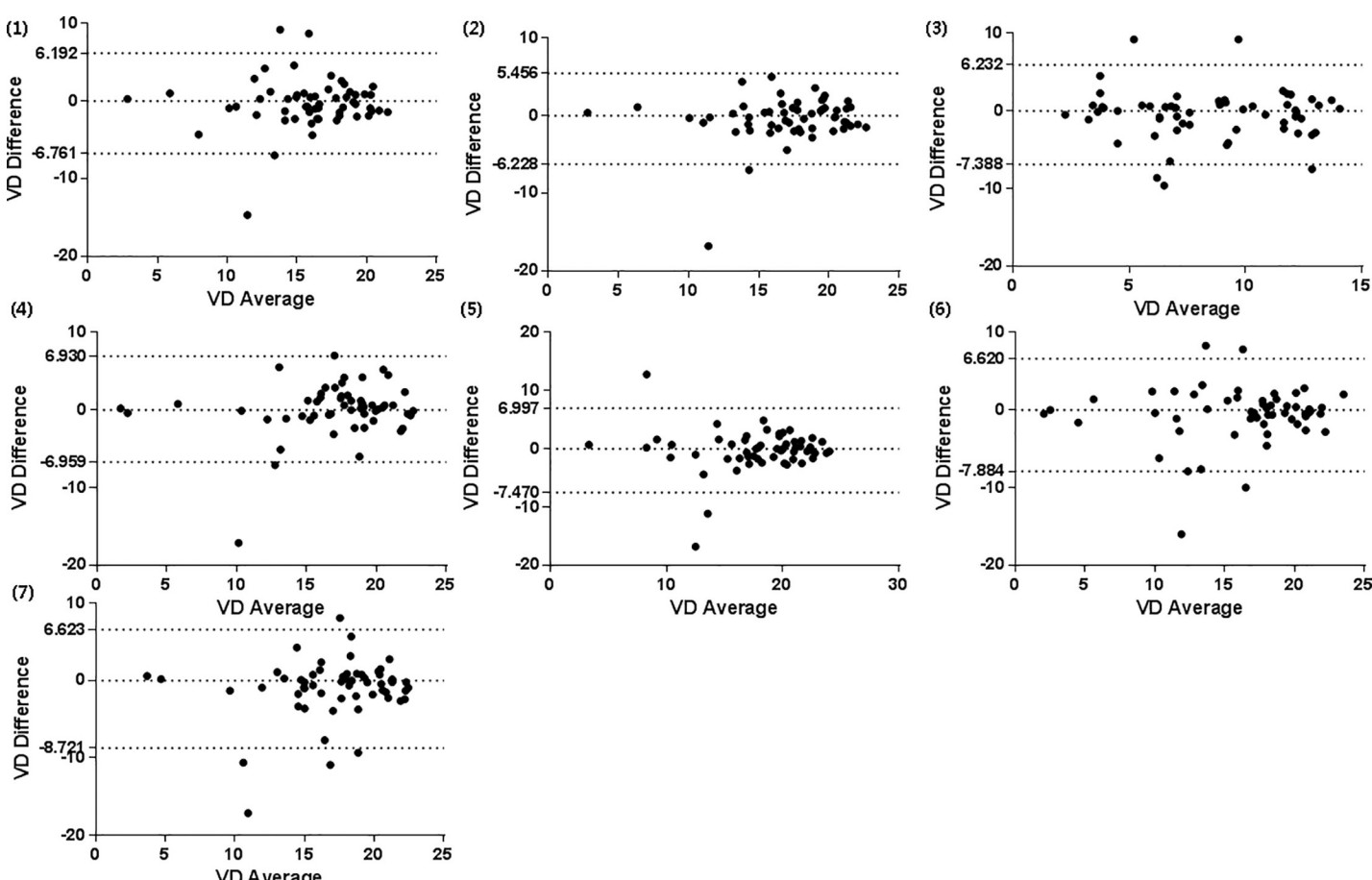

**Fig 1. Bland-Altman plots showing the level of agreement for the VD measurements obtained using AngioPlex optical coherence tomography between two consecutive measurements in patients with RVO.** VD measurements of 7 different areas: (1)Full area, (2) Inner area, (3)Central, (4)Superior, (5)Nasal, (6)Inferior, (7) Temporal. Two dot lines indicate the upper and lower boundaries of the 95% CIs. VD, vessel density.

**Table 4. Univariate and multivariate linear regression for the association between clinical and anatomical parameters and coefficient of variation of vessel density.**

| | Univariate | | Multivariate | |
|---|---|---|---|---|
| | B (95% CI) | P values | B (95% CI) | P values |
| Age | -0.002(0.003) | 0.577 | | |
| Sex | -0.012(0.043) | 0.778 | | |
| Laterality | -0.001(0.042) | 0.989 | | |
| Phakic eye | 0.010(0.061) | 0.865 | | |
| Diabetes | -0.009(0.048) | 0.849 | | |
| Hypertension | -0.061(0.048) | 0.204 | | |
| SE | -0.010(0.015) | 0.487 | | |
| BCVA | 0.053(0.081) | 0.519 | | |
| IOP | -0.002(0.008) | 0.833 | | |
| Axial length | -0.034(0.033) | 0.315 | | |
| Mean signal strength | 0.016(0.025) | 0.518 | | |
| Mean CMT | 0.001(0.000) | **0.003** | 0.001(0.000) | **0.002** |
| Mean GC-IPL thickness | -0.002(0.001) | **0.020** | 0.000(0.001) | 0.877 |

Values with p<0.05 are shown in bold.

BCVA, best corrected visual acuity; CMT, central macular thickness; GC-IPL, ganglion cell-inner plexiform layer; IOP, intraocular pressure; SE, spherical equivalent.

in 37 eyes. Among 37 eyes, the FAZ line was inappropriate in 13 eyes. FAZ line was drawn either too small or away from the fovea. Therefore we couldn't investigate the repeatability of the FAZ metrics in this study.

## Factors affecting the repeatability of VD measurements

Univariate analyses showed that the mean CMT (B: 0.001; $p < 0.001$) and mean GC-IPL thickness (B: -0.002; $p = 0.020$) were significant factors that affected the repeatability. Multivariate analyses of these two factors showed that only the mean CMT was a significant factor (Table 4). Fig 2 shows a scatterplot graph of the VD differences in the average CMT between two consecutive measurements. Eyes with large central macular thicknesses showed a relatively large difference compared to eyes with normal central macular thicknesses, which showed a small difference.

We also conducted a subgroup analysis by dividing the eyes into two groups according to a CMT of 400 μm. We separately analyzed the ICC, CV, and TRTSD in the two groups. As a result, the ICC and CV of the full VD in group 1 (CMT > 400 μm) were 0.348 and 22.55%, respectively. In group 2 (CMT < 400 μm), the ICC and CV of the full VD were 0.910 and 7.76%, respectively (Table 5).

## Discussion

Since its introduction, many studies have used OCTA to analyze eyes with RVO. Rispoli et al [14] reported that OCTA detected foveal avascular zone enlargement, capillary nonperfusion, microvascular abnormalities, and vascular congestion signs both in the superficial and deep capillary network in all eyes affected by RVO. Suzuki et al. [15] also reported that the visualization of microvascular abnormalities in eyes with macular edema associated with RVO was equal or better using OCTA compared to FA. Lee et al. [16] showed that the repeatability of VD measurements using OCTA in various retinal diseases, including RVO, was relatively good. However, to the best of our knowledge, no studies have been reported that analyzed the repeatability of OCTA measurements extensively and exclusively in eyes with RVO.

# Scatter plot

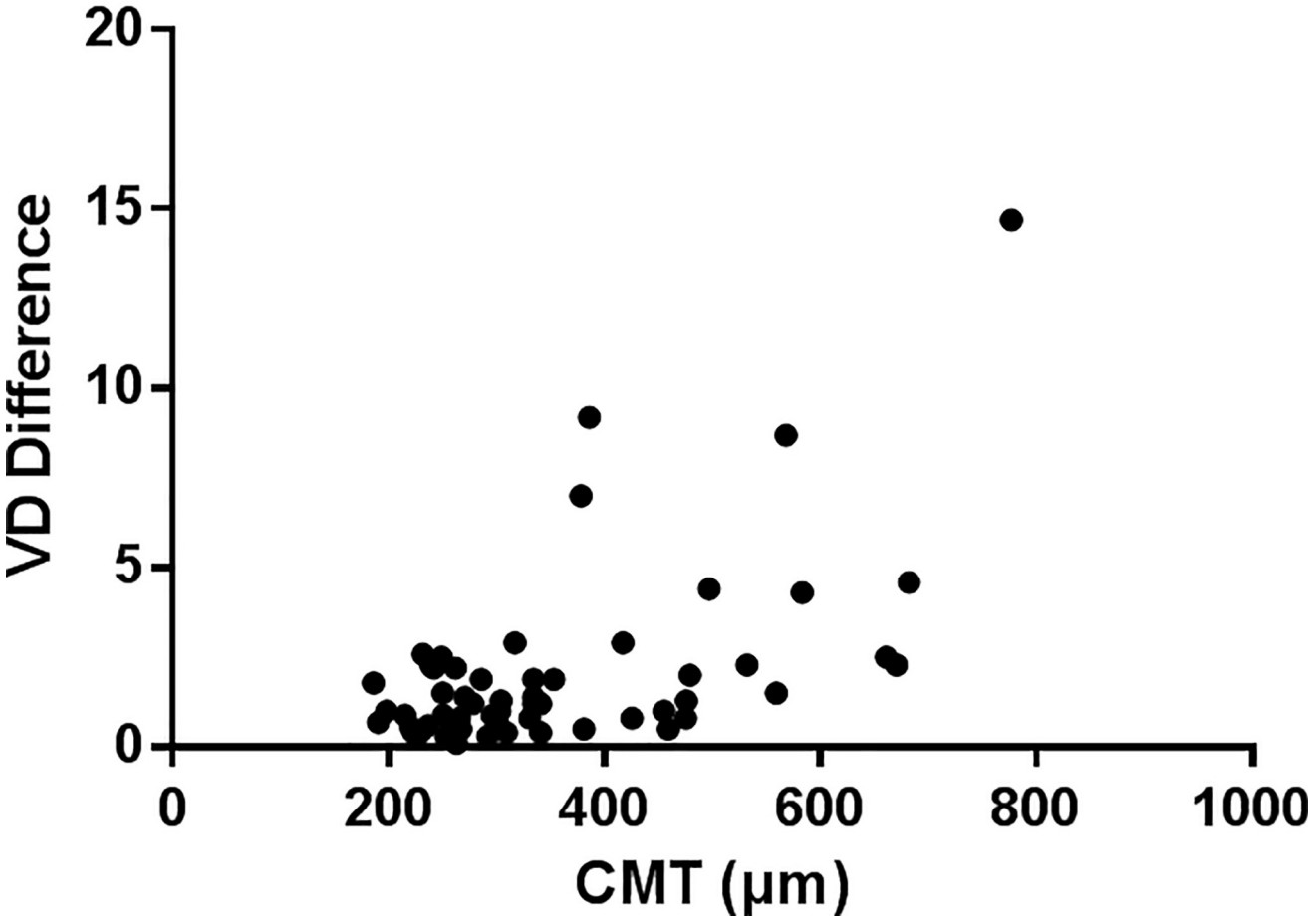

**Fig 2. Scatterplot graph of the differences in VD between two consecutive measurements using optical coherence tomography angiography.** The differences in VD tended to be larger when the CMT was thicker. CMT, central macular thickness; VD, vessel density.

Previous studies have reported good repeatability and reproducibility of OCTA in normal eyes [7–10]. Lee et al. [16] reported that the CV of the full VD was 10.65% in eyes with RVO, which was higher than those with other retinal diseases. In our study, the CV of the full VD was 10.61%, which was comparable. In both studies, the CV of the central area was higher than that of other areas. The reason is probably because the CV was affected by the average. The CV was calculated as $100 \times SD/overall$ mean, so the central area had a relatively lower VD than other areas. Additionally, Parodi et al. [17] reported an enlarged FAZ area in eyes diagnosed with RVO using FA. Other studies also reported an enlarged FAZ area using OCTA, consistent with previous findings [14, 15]. Therefore, a decreased VD measurement in the central area could be explained by an enlarged FAZ area (Table 3). This is perhaps the reason why the CV of the central area was higher than that of other areas. When analyzing the CV in eyes with RVO, we therefore need to consider the effect of an enlarged FAZ area, which can affect the VD average and the CV.

**Table 5. Subgroup analysis by CMT value of 400 μm.** First and second mean values, intraclass correlation coefficient, coefficient of variation and test-retest standard deviation of vessel density in patients with RVO.

| | | First mean VD | Second mean VD | ICC | CV (%) | TRTSD |
|---|---|---|---|---|---|---|
| CMT>400 μm | Full | 14.9±4.5 | 16.5±3.0 | 0.348 | 22.55 | 1.83 |
| | Inner | 16.0±4.5 | 17.3±3.2 | 0.439 | 13.47 | 1.35 |
| | Sectorial | | | | | |
| | Central | 9.36±3.9 | 11.2±3.1 | 0.235 | 29.24 | 1.82 |
| | Superior | 15.1±5.8 | 16.6±4.4 | 0.668 | 17.18 | 1.58 |
| | Nasal | 17.1±4.7 | 18.7±2.8 | 0.519 | 12.07 | 1.29 |
| | Inferior | 15.7±5.7 | 16.3±5.6 | 0.757 | 15.47 | 1.43 |
| | Temporal | 15.4±5.5 | 18.0±3.1 | 0.353 | 20.76 | 2.00 |
| CMT<400 μm | Full | 16.1±4.1 | 15.9±4.2 | 0.910 | 7.76 | 0.80 |
| | Inner | 17.2±4.3 | 17.2±4.2 | 0.946 | 6.26 | 0.71 |
| | Sectorial | | | | | |
| | Central | 7.83±3.5 | 7.79±3.6 | 0.863 | 18.46 | 0.81 |
| | Superior | 17.8±4.6 | 17.1±4.4 | 0.925 | 7.61 | 0.89 |
| | Nasal | 17.9±4.9 | 17.6±5.3 | 0.886 | 10.02 | 0.97 |
| | Inferior | 16.0±5.3 | 16.6±4.9 | 0.895 | 10.48 | 1.04 |
| | Temporal | 17.1±4.3 | 17.6±4.5 | 0.881 | 7.42 | 0.89 |

CV, coefficient of variation; ICC, intraclass correlation coefficient; TRTSD, test-retest standard deviation; VD, vessel density.

According to a univariate linear regression analysis, the mean CMT and GCIPL thickness were two factors affecting the repeatability of VD measurements. Lee et al. [16] suggested that in univariate linear regression analyses of the BCVA, the mean signal strength, mean CMT, and mean GCIPL thicknesses were significant factors affecting the repeatability of the VD measurement in various retinal diseases, including RVO, but a multivariate analysis showed that the GC-IPL thickness was the only factor. In the present study, GC-IPL thickness was also a significant factor using the univariate model. GC-IPL thickness was negatively correlated with the CV of VD measurements. The VD could have been decreased in eyes with a thinner GC-IPL, which may have caused the CV of VD measurements to increase. However, using multivariate analyses, the GC-IPL was not significant ($p = 0.877$). In eyes with macular edema, OCTA tends to measure a thinner GCIPL thickness than the actual thickness because of segmentation error [18, 19]. Therefore, the impact of the actual GCIPL thickness on the repeatability of OCTA might have been diminished; additional studies are needed to prove this hypothesis.

The only significant factor in the multivariate regression analyses was the mean CMT, which was negatively related with the repeatability of VD measurements. The mean CMT can affect VD measurements in three ways. First, in eyes with macular edema, the normal contour of the retinal layers is distorted and segmentation errors could easily occur, so it can significantly affect the VD measurements. In our study we discovered 7 eyes with definite segmentation error and all 7 eyes were affected by macular edema. Second, patients with macular edema had lower visual acuity. There might have been difficulties in achieving proper fixation when examining the eye. Lastly, macular edema can overshadow the retinal vasculature, which may also alter VD measurements and affect their repeatability. Interestingly, recent study by Nicolai et al suggest the application of OCTA on peripapillary area to evaluate microvascular changes when the CRVO is complicated with macular edema. According to their study, peripapillary metrics can be more reliable data when the macular metrics are unreliable due to macular edema [20].

To further examine the effect of CMT in VD measurements, we conducted a subgroup analysis dividing the eyes into two groups according to a CMT of 400 μm. Subgroup analysis

showed a distinct difference between the two groups in the repeatability of VD measurements. In group 1 (CMT > 400 μm), the ICC and CV were 0.348 and 22.55%, respectively, compared to 0.910 and 7.76%, respectively, in group 2. The repeatability was significantly lower in eyes with severe macular edema. Previously, many OCTA studies of RVO patients have included analyses of eyes with macular edema. These studies did not consider the effect of macular edema on VD measurements, which needs to be reconsidered because the OCTA measurements may have not been reliable [21, 22]. Also, clinicians should be aware of the effect of macular edema when interpreting OCTA data, especially when the CMT is > 400 μm.

BRVO can occur in different regions according to the site of the occlusion. We expected lower repeatability of VD measurements in the area involved with BRVO region and analyzed 35 eyes with BRVO. However, the CV of VD measurement in involved and uninvoled areas were 9.2% and 10.5% respectively, which did not show definite difference. The results can be explained by the effect of macular edema. In our study 26 eyes (74%) had macular edema and in most of the eyes with macular edema, all of the ETDRS inner circle area, which we analyzed in this study, were affected by the edema regardless of the BRVO region. Further studies analyzing OCTA scans with wider area are needed.

Foveal avascular zone enlargement in eyes with RVO is a well-known phenomenon. Parodi et al. [17] reported enlargement of FAZ in BRVO using FA and its high correlation with visual impairment. Other studies also reported similar FAZ changes using OCTA [14,15]. Therefore the reliability of OCTA in detecting FAZ is a important factor to consider when evaluating patients with RVO. In our study we couldn't analyze the repeatability of FAZ metrics. Angioplex software failed to detect FAZ in 20 eyes. Additionally, 13 eyes had inappropriate FAZ line that was drawn automatically by the software. The automated FAZ measurements weren't reliable and manual measurements would be neccessary when analyzing FAZ metrics in eyes with RVO.

There are several limitations to this study. We only used 3 × 3 mm scan images for analysis. Currently, more pattern scan settings are provided by the OCTA. However, according to a previous study, 3 × 3 mm scans obtained better image resolution than 6 × 6 mm scans when measuring the superficial retinal VD, making it more appropriate for analysis [23]. We could not analyze the VD of the deep retinal layer because the AngioPlex software detected only the VD of the superficial retinal layer. However, due to projection artifacts it is known that an analysis of the superficial retinal layer is more accurate than that of the deep retinal layer. We only scanned the OCTA twice per each eye to investigate repeatability due to the limitation of clinical situation. Finally, we could not analyze the data after manual segmentation because current version of AngioPlex software does not provide quantitative data when manual segmentation is performed. We need further study on the reliability of manual segmentation in OCTA.

In conclusion, OCTA is a practical tool for evaluating eyes with RVO, and it can be used for various clinical measurements. However, when analyzing the OCTA results in eyes with RVO associated with macular edema, clinicians should consider the possibility of its lower repeatability and reliability.

## Supporting information

**S1 Data.**
(XLSX)

## Author Contributions

**Conceptualization:** Min-Woo Lee, Jung-Yeul Kim.

**Data curation:** Kyeung-Min Kim, Hyung-Bin Lim, Hyung-Moon Koo, Jung-Yeul Kim.

**Formal analysis:** Kyeung-Min Kim.

**Methodology:** Min-Woo Lee.

**Supervision:** Min-Woo Lee, Yong-Il Shin, Jung-Yeul Kim.

**Visualization:** Hyung-Bin Lim, Hyung-Moon Koo.

**Writing – original draft:** Kyeung-Min Kim.

**Writing – review & editing:** Min-Woo Lee, Hyung-Bin Lim, Yong-Il Shin, Jung-Yeul Kim.

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
