## [Decision Letter · Decision Letter 0]

2 Apr 2020

PONE-D-20-06442

Repeatability of measuring the vessel density in patients with retinal vein occlusion: an optical coherence tomography angiography study.

PLOS ONE

Dear DR. Kim,

Thank you for submitting your manuscript to PLOS ONE. After careful consideration, we feel that it has merit but does not fully meet PLOS ONE’s publication criteria as it currently stands. Therefore, we invite you to submit a revised version of the manuscript that addresses the points raised during the review process.

We would appreciate receiving your revised manuscript by May 17 2020 11:59PM. To enhance the reproducibility of your results, we recommend that if applicable you deposit your laboratory protocols in protocols.io, where a protocol can be assigned its own identifier (DOI) such that it can be cited independently in the future. For instructions see: http://journals.plos.org/plosone/s/submission-guidelines#loc-laboratory-protocols

We look forward to receiving your revised manuscript.

Kind regards,

Ireneusz Grulkowski, PhD

Academic Editor

PLOS ONE

Reviewers' comments:

Reviewer's Responses to Questions

**Comments to the Author**

1. Is the manuscript technically sound, and do the data support the conclusions?

Reviewer #1: Partly

Reviewer #2: Yes

Reviewer #3: Yes

2. Has the statistical analysis been performed appropriately and rigorously? 

Reviewer #1: No

Reviewer #2: Yes

Reviewer #3: Yes

3. Have the authors made all data underlying the findings in their manuscript fully available?

Reviewer #1: No

Reviewer #2: Yes

Reviewer #3: Yes

4. Is the manuscript presented in an intelligible fashion and written in standard English?

Reviewer #1: No

Reviewer #2: Yes

Reviewer #3: Yes

5. Review Comments to the Author

Reviewer #1: Comments for the Authors: PONE-D-20-06442

1) General comments

The authors assessed the repeatability of the vessel density measurements by the OCTA, however, the authors did not clarify what have been investigated so far and what kind of unsolved research questions are going to be tested in the study. Also, the flow of the sentences is usually off and hard to understand the sales point of the manuscript.

Specific comments:

- Abstract: Please spell out several abbreviations (e.g. SD-OCTA, VD, CV etc.) at their first appearance in the abstract.

- Abstract: lease specify what the full VD means.

- Abstract: The terms "group 1" and "group 2" were suddenly come up in the Results. Please define those in the Methods properly.

- Introduction: The authors should reconsider the flow of sentences in the introduction, which do not sound well and hard to understand. For example, the introduction said that RVO were classified into two categories.., but this information is not related to the following sentence and would be redundant.

- Introduction: The authors stated OCTA has provided many benefits though there are many limitations in FA - however, the authors did not explain what are the "many benefits" of OCTA and not only pros but also their cons should be explained.

- Introduction: What is the sale point of the study? - Could not find any new information/investigations from the study. What is the strong point of the study, which has not been investigated?

- Introduction: The authors did not put appropriate citations throughout the introduction section. Please cite previous paper which support what the authors stated.

- Methods: Please provide the diagnosing criteria for RVO - did the atuhors use FA or just an OCTA?

- Methods: Methods: Did the authors choose one eye per patient or sometimes both eyes? If both eyes were used, specific stat models such as GLM/ GEE should be considered to employ.

- Methods: Demographics should be also shown by CMT category, i.e., <400 and >400 groups.

Reviewer #2: The manuscript by Dr. Kim et al investigated the repeatability of vessel density measurement in RVO patients and explore its related factor. It found that in eyes with macular edema, the repeatability is poor. Generally it is interesting, but further revisions are needed.

1. Please clarify how the sample size be calculated. This study only repeated the scan twice. Why didn't you scan more times?

2.This study found that macular edema is a risk factor of poor repeatability, it also give some explaination in the discussion. I would like to suggest the author to quantifiy segmentation error in their images and report the repeatability after manual correction of segmentation.

3. This study only reported the results of vessel density. The metrics of foveal avascular zone are also important paramters on OCTA. Please also investigate the repeatability of FAZ metrics.

4. The study subjects are mixed of CRVO and BRVO. There are also different region of BRVO. It would be interesting to investigate whether the repeatability is different between the region involved and not involved.

Reviewer #3: The authors present an assessment of OCTA repeatability in retinal vein occlusion.

The authors also report a linear regression analysis of clinical and anatomical parameters affecting VD repeatibility and demonstrate how the increase of macular volume reduce the accuracy of the examination.

Our group recently published an angio-OCT study to evaluate Papillary Vessel Density Changes after Intravitreal Anti-VEGF Injections in Central Retinal Vein Occlusion (J Clin Med. 2019 Oct 6;8(10). pii: E1636. doi: 10.3390/jcm8101636). We believe that the peripapillary area might be a region of interest to study microvascular changes when significant macular edema is present. I think it would be interesting to add a comment on this subject.

In the conclusions section it would be more appropriate to specify that there is a correlation between the entity of macular edema and the repeatability of VD measurement rather than stating that the repeatability is "poor when eyes were associated with macular edema".

Minor corrections:

Page 13 line line 120: typos thickness

Page 15 line 220 lower instead of low

6. PLOS authors have the option to publish the peer review history of their article (what does this mean?). If published, this will include your full peer review and any attached files.

Reviewer #1: No

Reviewer #2: Yes: Haoyu Chen

Reviewer #3: Yes: Michele Nicolai

---

## [Author Response · Author response to Decision Letter 0]

17 May 2020

Reviewer #1: Comments for the Authors: PONE-D-20-06442

1) General comments

The authors assessed the repeatability of the vessel density measurements by the OCTA, however, the authors did not clarify what have been investigated so far and what kind of unsolved research questions are going to be tested in the study. Also, the flow of the sentences is usually off and hard to understand the sales point of the manuscript.

: Thank you for your sincere review. We agree the flow of the manuscript seems unnatural and the sale point was mentioned vaguely. We have refined the sentences and emphasized the strong point of the study in the introduction. 

Specific comments:

- Abstract: Please spell out several abbreviations (e.g. SD-OCTA, VD, CV etc.) at their first appearance in the abstract.

- Abstract: Please specify what the full VD means.

- Abstract: The terms "group 1" and "group 2" were suddenly come up in the Results. Please define those in the Methods properly.

: Thank you for your comments. We have spelled out the abbreviations at their first appearance. We have explained the meaning of full VD in its first appearance in the abstract. We have added the definition of the subgroups in the method section in the abstract.

- Introduction: The authors should reconsider the flow of sentences in the introduction, which do not sound well and hard to understand. For example, the introduction said that RVO were classified into two categories.., but this information is not related to the following sentence and would be redundant.

: Thank you for your comment. We have read the Introduction thoroughly considering the flow of sentences. We have erased the sentence stating the classification of RVO that is thought to be redundant. Considering the flow we revised the sentences in the introductions. (Page 4, line 61-64, 65-68)

- Introduction: The authors stated OCTA has provided many benefits though there are many limitations in FA - however, the authors did not explain what are the "many benefits" of OCTA and not only pros but also their cons should be explained.

: Thank you for your comment. We have stated the benefits of OCTA in the introduction. “OCTA is noninvasive and less time-consuming, and it provides depth-resolved images to visualize the retinal vasculature in multiple layers. OCTA also provides quantitative metrics of the retinal microvasculature”. (Page 4, line 57-61) Even though we have stated the pros of the OCTA, for better understanding we have added the cons of OCTA in the following sentence. “Even though OCTA has several limitations such as projection artifacts and narrow field of view, this novel technique can provide clinicians with microvascular information that can assist diagnosis and treatment of many types of retinal vascular diseases.” (Page 4, line 61-64) 

- Introduction: What is the sale point of the study? - Could not find any new information/investigations from the study. What is the strong point of the study, which has not been investigated?

: We agree with your comment. We have stated the sale point of this study in the introductions but it seems ambiguous. For concise expression we have revised and added following sentences. ‘Many studies reported microvascular changes in eyes with RVO using OCTA. (3,14,15) At the same time, the reliability and efficacy of the OCTA have been questioned. The repeatability of this new device have been reported in many studies in normal eyes, however, there is limited study on the repeatability of OCTA in eyes with RVO.’ (Page 4, line 65-68) 

- Introduction: The authors did not put appropriate citations throughout the introduction section. Please cite previous paper which support what the authors stated.

: Thank you for your comment. We have added total of 15 references to support the statements made in the introduction section.

- Methods: Please provide the diagnosing criteria for RVO - did the authors use FA or just an OCTA?

: Thank you for your suggestion. We diagnosed RVO through funduscopy exam, OCT and FA findings. We have added the following statement in the method section (page 5, line 78-79)

- Methods: Methods: Did the authors choose one eye per patient or sometimes both eyes? If both eyes were used, specific stat models such as GLM/ GEE should be considered to employ.

: Thank you for your comment. We chose one eye per patient in this study. To avoid the confusion we have stated in the Demographics ‘A total of 57 eyes from 57 patients were examined’.(page 7, line 120-121)

- Methods: Demographics should be also shown by CMT category, i.e., <400 and >400 groups.

: Thank you for your comment. We have added table 2 for demographics of each subgroups. There were no significant characteristic differences between the two group except for CMT. We added the following statement in the demographics. (Page 7, line 124-126)

Reviewer #2: The manuscript by Dr. Kim et al investigated the repeatability of vessel density measurement in RVO patients and explore its related factor. It found that in eyes with macular edema, the repeatability is poor. Generally it is interesting, but further revisions are needed.

1. Please clarify how the sample size be calculated. This study only repeated the scan twice. Why didn't you scan more times?

: Thank you for your comment. Thank you for your comment. We did not perform sample size calculations in this study. We tried to include more than 50 eyes so that various variables including within subject standard deviation can follow the normal distribution. We have reviewed every clinical record from August 2017 to August 2018 retrospectively. Total of 62 patients were recruited for the study and 5 patients were excluded due to poor image quality. 

Most of the patients in this study were old aged and scanning multiple times can cause fatigue and dryness of the eyes. Therefore, we were limited to scan patients twice since it was a retrospective study based on clinical situation. We have added the following limitation in the discussion. 

Most of the patients in this study were old aged and scanning multiple times can cause fatigue and dryness of the eyes. Therefore, we were limited to scan patients twice since it was a retrospective study based on clinical situation. We have added the following limitation in the discussion. 

2. This study found that macular edema is a risk factor of poor repeatability, it also give some explanation in the discussion. I would like to suggest the author to quantify segmentation error in their images and report the repeatability after manual correction of segmentation. 

: Thank you for your suggestion. This study is aimed to investigate the factors that can affect the OCTA result by autosegmentation which we mostly encounter in the clinic.We have identified 7 eyes with definite segmentation errors and all eyes were affected by macular edema. We highly agree with your comment on the need for manual segmentation. However we couldn’t perform the manual segmentation because the current version of OCTA software cannot measure the VD when manual segmentation is performed afterwards. We have added the following information in the discussion and limitation. (Page 15, line 225-226, page 17 266-268)

3. This study only reported the results of vessel density. The metrics of foveal avascular zone are also important parameters on OCTA. Please also investigate the repeatability of FAZ metrics. 

: Thank you for your suggestion. We also think that FAZ metrics are important in evaluating RVO. However, we had to exclude 20 eyes because Angioplex software failed to detect FAZ. And among the 37 eyes which Angioplex drew the FAZ line automatically, 13 eyes had inappropriate FAZ line that was either too small or away from the fovea. We drew the conclusion that automated FAZ analysis is unreliable when analyzing eye with RVO. We added in the result (Page 14, line 152-156) and also in discussion (Page 16-17, line 250-257). 

4. The study subjects are mixed of CRVO and BRVO. There are also different region of BRVO. It would be interesting to investigate whether the repeatability is different between the region involved and not involved.

: Thank you for your suggestion. We reviewed the OCTA images and metrics of 35 eyes with BRVO. We compared the OCTA images of the regions that were involved and not involved. However, there was no definite difference between the two group. In our study 26 eyes(74%) had macular edma and in most of the eyes with macular edema, all of the ETDRS inner circle area, which we analyzed in this study, were affected regardless of the BRVO region. Therefore the analysis according to the BRVO region was limited. We discussed the following matter in the discussions. (Page 16, line 243-250) 

Reviewer #3: The authors present an assessment of OCTA repeatability in retinal vein occlusion.

The authors also report a linear regression analysis of clinical and anatomical parameters affecting VD repeatability and demonstrate how the increase of macular volume reduce the accuracy of the examination.

Our group recently published an angio-OCT study to evaluate Papillary Vessel Density Changes after Intravitreal Anti-VEGF Injections in Central Retinal Vein Occlusion (J Clin Med. 2019 Oct 6;8(10). pii: E1636. doi: 10.3390/jcm8101636). We believe that the peripapillary area might be a region of interest to study microvascular changes when significant macular edema is present. I think it would be interesting to add a comment on this subject.

: Thank you for your thoughtful suggestion.When macular metrics are unreliable due to macular edema, peripapillary metrics can be more reliable and significant data to investigate retinal microvascular changes as you pointed out. We have added the comment in our discussion. (Page 15-16, line 229-232)

In the conclusions section it would be more appropriate to specify that there is a correlation between the entity of macular edema and the repeatability of VD measurement rather than stating that the repeatability is "poor when eyes were associated with macular edema".

: Thank you for your suggestion. We have changed the expression in the conclusions section. ‘Repeatability of VD measurement was significantly affected by central macular thickness.’ ( Page 3, line 46-47) 

Minor corrections:

Page 13 line line 120: typos thickness

Page 15 line 220 lower instead of low

: Thank you for your comment. We have checked and corrected the typos.

---

## [Decision Letter · Decision Letter 1]

5 Jun 2020

Repeatability of measuring the vessel density in patients with retinal vein occlusion: an optical coherence tomography angiography study.

PONE-D-20-06442R1

Dear Dr. Kim,

We’re pleased to inform you that your manuscript has been judged scientifically suitable for publication and will be formally accepted for publication once it meets all outstanding technical requirements.

Kind regards,

Ireneusz Grulkowski, PhD

Academic Editor

PLOS ONE

Additional Editor Comments (optional):

Reviewers' comments:

Reviewer's Responses to Questions

**Comments to the Author**

1. If the authors have adequately addressed your comments raised in a previous round of review and you feel that this manuscript is now acceptable for publication, you may indicate that here to bypass the “Comments to the Author” section, enter your conflict of interest statement in the “Confidential to Editor” section, and submit your "Accept" recommendation.

Reviewer #2: All comments have been addressed

Reviewer #3: All comments have been addressed

2. Is the manuscript technically sound, and do the data support the conclusions?

Reviewer #2: Yes

Reviewer #3: Yes

3. Has the statistical analysis been performed appropriately and rigorously? 

Reviewer #2: Yes

Reviewer #3: Yes

4. Have the authors made all data underlying the findings in their manuscript fully available?

Reviewer #2: Yes

Reviewer #3: Yes

5. Is the manuscript presented in an intelligible fashion and written in standard English?

Reviewer #2: Yes

Reviewer #3: Yes

6. Review Comments to the Author

Reviewer #2: (No Response)

Reviewer #3: The authors have addressed the comments. The English appears correct. The conclusion is now more in accordance with the findings of the study. I think the study provides interesting data on the repeatability of OCTA assessment in patients with macular edema correlated to RVO.

7. PLOS authors have the option to publish the peer review history of their article (what does this mean?). If published, this will include your full peer review and any attached files.

Reviewer #2: No

Reviewer #3: No

---

## [Editor Report · Acceptance letter]

12 Jun 2020

PONE-D-20-06442R1 

Repeatability of measuring the vessel density in patients with retinal vein occlusion: an optical coherence tomography angiography study. 

Dear Dr. Kim:

I'm pleased to inform you that your manuscript has been deemed suitable for publication in PLOS ONE. Congratulations! Your manuscript is now with our production department. 

Kind regards, 

on behalf of

Dr. Ireneusz Grulkowski 

Academic Editor

PLOS ONE